# Boosting Acetylcholine Signaling by Cannabidiol in a Murine Model of Alzheimer’s Disease

**DOI:** 10.3390/ijms252111764

**Published:** 2024-11-01

**Authors:** Hesam Khodadadi, Évila Lopes Salles, Sahar Emami Naeini, Bidhan Bhandari, Hannah M. Rogers, Jules Gouron, William Meeks, Alvin V. Terry, Anilkumar Pillai, Jack C. Yu, John C. Morgan, Kumar Vaibhav, David C. Hess, Krishnan M. Dhandapani, Lei P. Wang, Babak Baban

**Affiliations:** 1Department of Neurology, Medical College of Georgia, Augusta University, Augusta, GA 30912, USA; hchamgordani@augusta.edu (H.K.); jmorgan@augusta.edu (J.C.M.); dhess@augusta.edu (D.C.H.); 2Department of Oral Biology and Diagnostic Sciences, Dental College of Georgia, Augusta University, Augusta, GA 30912, USA; esalles@augusta.edu (É.L.S.); semaminaeini@augusta.edu (S.E.N.); bibhandari@augusta.edu (B.B.); harogers@augusta.edu (H.M.R.); gouronjules@gmail.com (J.G.); lewang@augusta.edu (L.P.W.); 3Center for Excellence in Research, Scholarship and Innovation (CERSI), Dental College of Georgia, Augusta University, Augusta, GA 30912, USA; 4The Graduate School, Augusta University, Augusta, GA 30912, USA; 5Medical College of Georgia, Augusta University, Augusta, GA 30912, USA; 6Department of Pharmacology and Toxicology, Medical College of Georgia, Augusta University, Augusta, GA 30912, USA; aterry@augusta.edu; 7Translational Psychiatry Program, Faillace Department of Psychiatry and Behavioral Sciences, The University of Texas Health Science Center at Houston (UTHealth), Houston, TX 77030, USA; anilkumar.r.pillai@uth.tmc.edu; 8Department of Surgery, Medical College of Georgia, Augusta University, Augusta, GA 30912, USA; jyu@augusta.edu; 9Department of Neurosurgery, Medical College of Georgia, Augusta University, Augusta, GA 30912, USA; kvaibhav@augusta.edu (K.V.); kdhandapani@augusta.edu (K.M.D.)

**Keywords:** acetylcholine, cannabidiol, CBD, Alzheimer’s disease, innate lymphoid cells, ILC2

## Abstract

Alzheimer’s disease (AD) is a challenging medical issue that requires efficacious treatment options to improve long-term quality of life. Cannabidiol (CBD) is a cannabis-derived phytocannabinoid with potential health benefits, including reports from our laboratory and others showing a therapeutic role in the pre-clinical treatment of AD; however, the mechanisms whereby CBD affects AD progression remain undefined. Innate lymphoid cells (ILCs) are recently discovered immune cells that initiate and orchestrate inflammatory responses. ILC2, a sub-class of ILCs, is proposed to have a role in cognitive function via unknown mechanisms. In this present study, we explored whether CBD ameliorates AD symptoms via the enhancement of acetylcholine (ACh), a cholinergic neurotransmitter involved in cognition that may regulate ILC2. 5xFAD mice were chronically treated by inhalation of a formulation of broad-spectrum CBD for seven months. ACh production, ILC2s profile, brain histopathology, and long-term behavior were assessed. Together, our studies showed that long-term inhalation of CBD improved cognitive function and reduced senile plaques in a murine AD model, effects that were associated with enhanced ACh production and altered ILC2s distribution within the CNS. These findings indicate that inhaled CBD could offer a cost-effective, non-invasive, and effective treatment for managing AD. The beneficial effects of CBD inhalation may be linked to increased ACh production and an altered distribution of ILC2s, highlighting the need for further research in this area.

## 1. Introduction

Alzheimer’s disease (AD) is a multifaceted disorder characterized by progressive and irreversible cognitive deterioration, which imposes a significant emotional and financial strain on society [1]. Often referred to as the “Silver Tsunami” of the 21st century, if the burden continues to grow at the same rate as in the past decade, AD is projected to impact over 150 million people worldwide with a financial encumbrance of $1.1 trillion by 2050, becoming the most challenging and expensive medical disorder globally. Although there have been gradual improvements in the treatment and understanding of Alzheimer’s disease (AD) pathophysiology, no definitive therapy exists to prevent or slow its progression [2,3]. Current treatments address only the symptoms rather than the complex underlying causes, leading to incomplete and ineffective management of the disease. Recent medications for AD, such as aducanumab and lecanemab, have received approval from the Food and Drug Administration (FDA) and show promising therapeutic potential. However, their long-term side effects and overall efficacy are still being investigated [4].

Cannabidiol (CBD) is a relatively safe, non-psychoactive phytocannabinoid produced by cannabis plants. Recent work carried out at our laboratory and others suggests a beneficial effect of CBD, either alone or in combination with other cannabinoids, in pre-clinical models of AD and neurodegeneration [1,5]. In particular, we reported that short-term administration of CBD alleviated the symptoms of AD, including reduced β-amyloid peptide (Aβ) deposition, in 5xFAD mice via modulation of TREM2 and IL-33 [1]. However, the molecular and cellular mechanisms underlying the ability of CBD to slow neurodegenerative processes are largely unknown. Moreover, the efficacy of chronic inhalant CBD remains unexplored as a therapy for AD.

The interplay between the nervous and immune systems is critical for the maintenance of physiological homeostasis [6,7]. In particular, the cholinergic system orchestrates bi-directional crosstalk between the nervous system and immunologic components. The cholinergic system consists of neurotransmitters, receptors, and enzymes that dynamically work in concert to transmit signals from the central and peripheral nervous systems to the immune network [8,9]. Acetylcholine (ACh), the most abundant neurotransmitter, is composed of an ester linking acetic acid and choline [10,11]. ACh regulates a diverse set of physiologic processes, including muscle contraction, sensory gating, arousal, learning, attention, and memory [10,11,12]. In fact, the introduction of the “cholinergic hypothesis” of Alzheimer’s disease (AD) was based on ACh deficiency as a way to explain the excessive neuroinflammation, extracellular deposition of β-amyloid (Aβ), and irregular phosphorylation of tau protein, all of which affect cognitive function [13,14]. Given the excitatory nature of ACh, any disturbance in cholinergic signaling may profoundly and irreversibly affect neuro-immune homeostasis [15], accelerating cognitive demise and earlier onset dementia [16,17]. Thus, targeted modulation of ACh may provide an attractive therapeutic approach to ameliorate the negative consequences of AD [18,19,20].

Although mainly produced by cholinergic neurons, immune cells of lymphocytic lineage, including B and T cells, and innate lymphoid cells (ILCs), may also produce ACh [21,22,23,24]. ILCs are functionally diverse immunomodulatory cells with crucial roles in the initiation and development of inflammatory responses, as well as tissue remodeling and homeostasis [25,26]. ILCs are tissue-resident cells, usually functionally quiescent and in a state of homeostasis, characterized by several unique cellular features that distinguish them from other lymphocytes and myeloid cells [27,28]. ILCs, which are divided into three subgroups (ILC1, ILC2, and ILC3) based on phenotype and function, lack known lineage markers, do not express antigen-specific B and T cell receptors, and exhibit a lower frequency than classical adaptive lymphocytes [29,30]. ILCs strategically reside at tissue gateways and barriers, rapidly responding to dynamic stimuli to orchestrate and coordinate immune responses to distinct challenges [25,31].

Several studies have indicated that ILC2s play a significant role in improving cognitive function and alleviating pathologic age-related symptoms [32,33]. Further, large accumulations of IL-5/IL-13-expressing ILC2s in the choroid plexus of aged mice have already been reported [32,34]. The induction of IL-5 and IL-13 by ILC2s has been shown to regulate neuroinflammatory responses, improving symptoms of age-related cognitive decline [33,35]. Such a protective role suggests that ILC2s could be a potential target as an immunotherapeutic modality in the treatment of several age-related disorders and deserves further investigation.

Herein, we showed that long-term cognitive benefits of inhalant CBD were associated with enhanced ACh production and changes in the profile of ILCs in a mouse model of AD, suggesting an immune-cholinergic approach in the treatment of AD.

## 2. Results

### 2.1. Long-Term Treatment with Inhaled CBD Increased Expression of Hippocampal ACh in AD Mice

Long-term treatment with inhaled CBD for seven months induced brain ACh expression in 5xFAD mice compared to those given the placebo treatment (Figure 1a). Hippocampal ACh expression in CBD-treated mice was significantly higher than ACh expression in AD mice treated with placebo (*p* > 0.05). ACh expression in all experimental groups was quantified using the ImageJ software version 1.53e (Figure 1b,c).

### 2.2. Inhalant CBD Improved Cerebral Blood Flow in AD

Photoacoustic imaging of the brain demonstrated that inhalant CBD improved the cerebral blood flow significantly in 5xFAD mice compared to the placebo-treated group (Figure 2). The visual overview of blood flow was obtained by color doppler and was based on the color intensity.

### 2.3. CBD-Induced ACh Was Associated with Improvement in Cognitive Function in AD

A seven-month treatment with inhaled CBD improved cognitive function in 5xFAD mice compared to placebo-treated mice (Figure 3). The NOR test suggested an improvement in cognitive function in CBD-treated mice, evidenced by more time spent on exploration compared to the placebo group (DI increased to 0.4 ± 0.9 from −0.25 ± 0.7, *p* ≤ 0.03) (Figure 3a,b). Similarly, the OF test indicated differences in spatial distribution of activity, with the CBD-treated 5xFAD group showing increased time in the central zone (260 ± 70 s) compared to placebo-treated mice (180 ± 80 s, *p* ≤ 0.05) (Figure 3c,d).

### 2.4. Long-Term Inhaled CBD Reduced Senile Plaques in AD

The Bielschowsky silver stain demonstrated that treatment with inhaled CBD reduced the occurrence of neurofibrillary tangles and senile plaques (amyloid plaques) compared to placebo-treated 5xFAD mice (Figure 4). Senile plaques are extracellular deposits of the amyloid beta (Aβ) protein, and their reduction in CBD-treated subjects was consistent with improvements in cognitive function.

### 2.5. Inhaled CBD Changed the Profile of ILC2s in the Meninges and Choroid Plexus of 5xFAD Mice and Their Level of ACh Expression

Long-term treatment with inhaled CBD had profound impacts on the immunologic profile of brain of 5xFAD mice. CBD treatment reversed the frequency of meningeal ILC2s towards a level closer to WT mice, significantly higher than in the meninges of 5xFAD mice treated with placebo (*p* < 0.04) (Figure 5a–c). Concurrently, CBD treatment reduced the frequency of ILC2s in the CP of 5xFAD mice, significantly lower than that in placebo-treated mice (** *p* ≤ 0.02), and closer to the level in WT mice (Figure 6a–c). These novel findings are important since ILC2s are playing crucial roles in shaping brain function, orchestrating neuroinflammation, and optimizing cognitive function. More importantly, long-term treatment of 5xFAD mice with inhaled CBD increased ACh production in their meningeal ILC2s significantly compared to their counterparts treated with placebo (Figure 5d and Figure 6d). Given the nature of ACh as a signaling molecule, its expression by ILC2s could suggest a major regulatory mechanism in containing the excessive neuroinflammatory responses in AD. Additionally, the UMAP analysis showed a profound alteration in the distribution pattern of ILC2s in both meninges and the CP after CBD treatment compared to the placebo-treated 5xFAD mice (Figure 5e and Figure 6e).

## 3. Discussion

The findings presented here in this current study are significant and novel at several levels. This is the first report to suggest that the interplay between CBD and the cholinergic system could be crucial for the beneficial effects of CBD in the treatment of AD in a pre-clinical setting. We had previously shown that CBD ameliorated the adversarial symptoms and improved cognitive function in a murine model of AD [1]. The exact mechanisms by which CBD affects the pathophysiology of AD are still not completely understood [36]. However, various studies indicate that CBD may offer symptomatic relief or potentially slow the progression of AD through its anti-inflammatory, antioxidative, and neurogenic properties [36]. At the molecular level, CBD is recognized as an inverse agonist of cannabinoid receptors, and this modulation has been shown to be beneficial in preventing AD-related pathology [36]. Importantly, it is proposed that long-term treatment with CBD may ameliorate the AD symptoms through the upregulation of mRNA levels and its role in the amendment and optimization of autophagy in AD [36]. Interestingly, it is shown that activation of the alpha7 nicotinic acetylcholine receptor (a7 nAChR) can enhance autophagy and contributes to neuronal survival [37]. Therefore, it is plausible to hypothesize that long-term CBD inhalation increases ACh levels and its receptors, leading to the modulation of impaired autophagy in AD. Further, it is well established that a deficiency of ACh in patients with AD results in a gradual and considerable decline in cognitive and behavioral functions [19]. Our results demonstrated that long-term CBD treatment was able to increase ACh levels in a murine model of AD with no additional complications. Thus, it is rational to propose that CBD could provide a safe, natural, and affordable booster of ACh to improve the cognitive function and to achieve better outcomes in the treatment of AD, and warrants further investigations.

Increasing evidence accentuate that ACh is a key molecule in the regulation of immunoinflammatory responses and plays a crucial role in maintaining homeostasis [38]. In this study, we showed for the first time that long-term inhalation of broad-spectrum CBD induced ACh expression within the hippocampus and meningeal ILC2, paralleling reduced amyloid accumulation and cognitive improvements in 5xFAD mice. Given the role of ACh in regulating the inflammation [38], our results support the notion that targeting the cholinergic system provides an effective therapeutic modality in the treatment of AD and other neurodegenerative diseases. Enhancement of ACh expression by CBD maybe an important factor by which CBD could modulate the neuroinflammation within CNS and to mitigate the adversarial symptoms of AD.

Regulation of the cholinergic network may provide a mechanism to explain how CBD exerts potent anti-inflammatory effects in AD. ILCs maintain homeostasis and regulate the initiation and development of inflammatory responses during injury and infection. Of the three sub-classes of ILCs, ILC2s are specifically associated with cognitive function [32,39]. Notably, ILC2s accumulate within the choroid plexus during neuropathology and in the late stages of natural aging [32,33,40], while the cerebral meninges are a major residential site for ILC2s in younger ages and under healthy status [29,34]. The shift in location and accumulation of ILC2s from meninges to the CP during the aging process is associated with cognitive deficiency and impairment [32,33,41]. Our findings indicate that inhalation of CBD reverses the depletion of ILC2s from meninges and normalizes ILC2 numbers within the choroid plexus, which may calibrate the function of the choroid plexus and meninges, re-establish immune balance, and improve cognitive function. Given the beneficial effects of ACh in the treatment of AD, our observation that inhaled CBD increased ACh-expressing ILCs may provide a framework for development of a mechanistically innovative immunotherapeutic for the treatment of AD.

Moreover, inhalation as a mode of drug delivery has a number of advantages. Inhaled CBD is non-invasive, easy to apply, highly accurate with respect to dosing and target engagement, affordable, and readily available. Our studies suggest long-term CBD inhalation is both safe and efficacious in a murine AD model; however, further studies are needed in other species and models to establish whether CBD may exhibit high translational value.

### Study Limitations

This study has potential limitations. The expression of ACh in different brain areas, timing/stages of disease, CBD dosing/formulations, gender effect, and profile of different sub-classes of ILCs are examples of such limitations, and warrant further research. Thus, a more thorough investigation is essential to assess the safety, pharmacokinetics, pharmacodynamics, and, most critically, the efficacy of cannabinoid-based drugs in the treatment of AD.

## 4. Materials and Methods

### 4.1. Declaration Regarding Humane Use of Animals

All mouse experiments were conducted in accordance with the principles of the ‘Three Rs’ (Replacement, Reduction, and Refinement) and were approved by the Institutional Animal Care and Use Committee (IACUC) at Augusta University (Augustsa, GA, USA). Procedures were designed to minimize discomfort and distress to the animals, and all animal care practices adhered to the highest standards of humane treatment as outlined by the NIH (National Institute of Helath, Bethesda, Maryland, USA) guidelines.

### 4.2. Experimental Design and Treatment Protocol

Adult male 5xFAD mice were purchased from Jackson Laboratory (Bar Harbor, Maine, USA). 5xFAD mice express human amyloid precursor protein (APP) and preseniln-1 (PSEN1) transgenes with five AD-linked mutations, used as a pre-clinical model of AD [1]. At five months, mice were randomized into two groups (n = 10/group), receiving either placebo or inhaled CBD (10 mg/mouse, Thriftmaster Global Bioscience, Dallas, TX, USA) every other day for seven consecutive months. Each CBD inhaler contained 985 mg of broad-spectrum CBD (winterized crude hemp extract) plus 15 mg of co-solvent, surfactant, and propellant, with a total volume of 1000 mg (5 mg dose per actuation, with 200 mL/min flow rate). For the placebo, the 985 mg of broad-spectrum CBD was replaced with 985 mg of hemp seed oil. As described previously [42], inhalers were modified by adding an extra nozzle piece to adjust to the mouse model and to further control the intake of CBD.

### 4.3. Photoacoustic Imaging of Brain

Cerebral blood flow was measured using a FujiFilm VEVO 3100 imaging system. Ultrasound imaging (FUJIFILM VisualSonics Corporation, Bothell, WA, USA) was used to determine the morphologic regularity and blood flow in brain tissues, measured using the laser photoacoustic method. The key imaging parameters were frequency, dynamic range, depth of field, gain setting, and frame rate. After scanning, multiple frames of ultrasound data were captured to create a comprehensive view of the anatomical structures. Advanced algorithms were employed to enhance image quality, improve contrast, and reduce noise, providing clearer visuals for analysis.

### 4.4. Behavioral Tests

Open Field (OF) and Novel Object Recognition (NOR) tests were used to assess cognitive outcomes as described previously [1]. For NOR testing, mice were placed in an enclosed box with two identical objects that were placed within a 10 cm circle, at a set distance apart. Mice were then removed from the environment for a predetermined amount of time, and one of the two previously used (familiar) objects was replaced with a novel object that was different from the familiar object in shape, texture, and appearance. The ability of the mouse to discriminate between the familiar and novel object was quantified as a discrimination index, DI = (Tn − Tf)/(Tn + Tf), where Tn is the time spent by the mouse with the novel object, and Tf indicates the time spent with the familiar object. In OF evaluation, mice were tested in a square box (40 cm by 40 cm by 40 cm) for 10 min, and activity was digitally recorded. Distance traveled, mean velocity, and time spent in the center zone were analyzed with Ethovision XT video-tracking 17.5 software (Noldus Information Technology, Leesburg, VA, USA).

### 4.5. Immunohistochemistry

ACh immunostaining was performed, as previously described [1,42]. Briefly, 5 μm formalin-fixed and paraffin-embedded brain tissue sections were mounted onto glass slides and incubated with a specific antibody against ACh (LS-C295819, LSBio, Seattle, WA, USA), and incubated at 4 °C overnight. Preparations were counterstained with hematoxylin and mounted in Faramount. Slides were imaged by a blinded investigator using bright-field microscopy and quantified using the ImageJ software (version 1.53e).

### 4.6. Bielschowsky Silver Staining

Freshly harvested brain tissue was fixed in formalin and embedded in paraffin. Serial 8 μm coronal brain sections were cut using a sliding microtome and then sections were deparaffinized prior to staining with a Bielschowsky Stain Kit (Polysciences, Warrington, PA, USA, cat# 25994-250). Slides were rinsed in tap water, fixed in 5% sodium thiosulfate, dehydrated through alcohols and xylene, treated with ammonical silver, and then reduced to visible metallic silver to label senile plaques, axons, and neurofibrillary tangles. Slides were imaged by a blinded investigator with the use of bright-field microscopy, and quantified using the ImageJ software (version 1.53e).

### 4.7. Analytical Flow Cytometry

Meninges and choroid plexuses were carefully harvested, processed, and analyzed by flow cytometry as previously described [29,42]. ILC2s were identified as Lin^−^CD45^+^CD127^+^GATA3^+^; with expression of Interleukin-5 (IL-5) and IL-13. All antibodies from BioLegend (San Diego, CA, USA) unless otherwise noted. Cells were then run through a NovoCyte Quanteun flow cytometer (Agilent Technologies, Santa Clara, CA, USA). Cells were gated based on forward and side scatter properties and on marker combinations to select viable cells of interest. Single stains were used to set compensation, and isotype controls were used to determine the level of nonspecific binding. Quantified measurement was performed using FlowJo (version 11.0) analytical software (FlowJo LLC, Ashland, OR, USA). Further analysis was performed by measuring ACh expression by ILC2s (LS-C691651, LSBio, Seattle, WA, USA). As previously described [43], the FlowJo plugin for the algorithm “uniform manifold approximation and projection” (UMAP) was used to perform and display the pattern distribution of meningeal ILC2s.

### 4.8. Statistical Analysis

Brown-Forsythe and Welch ANOVA were employed to determine significance (*p* < 0.05) among the groups and for statistical analysis. These methods assessed not only the equality of group means but also adjusted the denominator of the F ratio to ensure it had the same expected value as the numerator.

## 5. Conclusions

The new findings from this study suggest a possible mechanism connecting the immuno-protective effects of CBD in AD to the expression of ACh in the cholinergic system. These compelling results indicate that, by enhancing ACh expression as a crucial regulatory molecule, CBD helps restore homeostasis by modulating inflammatory responses driven by ILC2s. The interaction between CBD and ACh influences the frequency, distribution, and function of ILC2s in the meninges and the choroid plexus, proposing a novel neuro-immunotherapeutic role for CBD as a safe and cost-effective treatment for Alzheimer’s disease.

### Clues for Future Directions

Although our new findings indicate promising CBD-based therapeutic approaches for AD, they also highlight the intricate interactions between CBD and ACh in the central nervous system, particularly during the onset and progression of the disease. Thus, it is crucial to conduct additional mechanistic studies to explore the specific biochemical pathways through which CBD affects ACh levels and cholinergic function, including receptor interactions and subsequent signaling cascades. Furthermore, there is an urgent need for clinical trials to assess the safety and efficacy of CBD in AD patients, with a focus on its effects on cognitive decline, behavioral symptoms, and ACh-related biomarkers. Only through these clinical trials can objective measures of success and potential adverse effects be assessed.

## Figures and Tables

**Figure 1 ijms-25-11764-f001:**
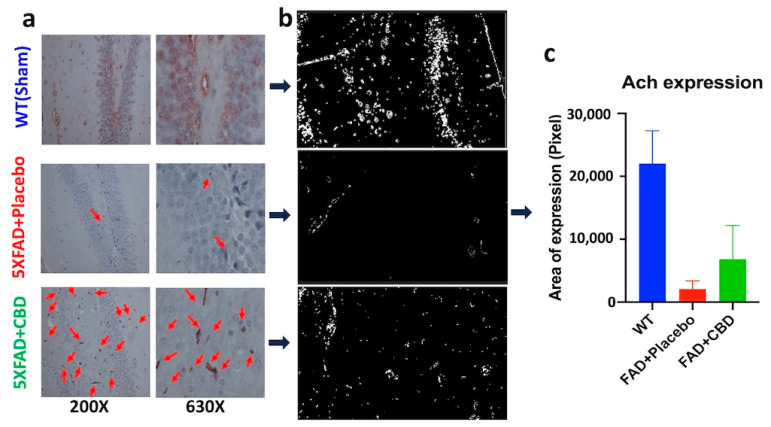
Increased acetylcholine induction by CBD inhalation in Alzheimer’s disease. CBD inhalation over a period of seven months (with dosing every other day) enhanced ACh expression in 5xFAD mice. (**a**) Immunohistochemical analysis on brain tissue from 5xFAD mice showed that long-term inhalation of CBD (seven months) increased ACh expression (red arrows) in 5xFAD mice with AD compared to placebo-treated mice. (**b**) Demonstration of ACh expression level in brain tissues using ImageJ software (version 1.53e) and (**c**) Quantification of ACh expression showing significant differences between ACh expression in brain tissue from AD mice treated with inhaled CBD versus placebo-treated mice. Level of ACh expression in normal WT mice was significantly higher than AD mice treated with either CBD or placebo.

**Figure 2 ijms-25-11764-f002:**
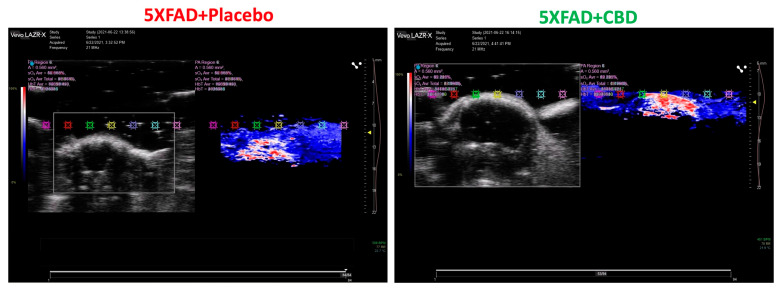
Inhaled CBD increased cerebral blood flow in mice with AD. Figure 2 illustrates cerebral perfusion and blood flow rate by measuring the percentage of oxyhemoglobin using a Laser Doppler Ultrasound machine. In this method, a laser perfusion scan quantifies the amount of oxyhemoglobin in the cerebral vasculature, providing a precise measurement of the cerebral perfusion rate. As shown in this figure, CBD treatment improves cerebral perfusion, suggesting a potential avenue for further investigation into how it may influence the speed of cognitive decline in vascular dementia and Alzheimer’s disease, conditions characterized by reduced cerebral perfusion.

**Figure 3 ijms-25-11764-f003:**
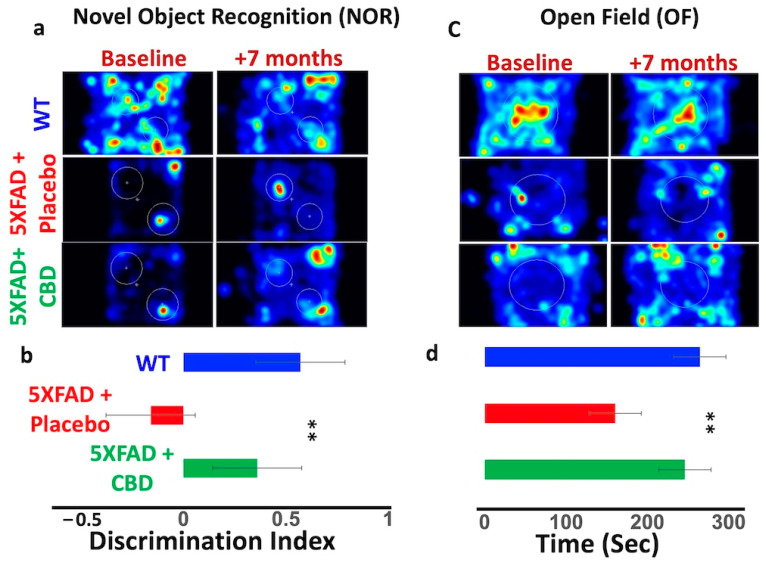
Long-term CBD inhalation improved cognitive function and ameliorated the AD symptoms. (**a**) Novel Object Recognition (NOR) test showing inhaled CBD improved cognitive function in 5xFAD mice compared to placebo-treated mice. Representative heat maps showing spatial distribution of mouse exploration in the testing arena, with warmer colors (e.g., red) indicating areas where the mouse spent more time exploring, and cooler colors (e.g., blue) indicating areas of less exploration. The familiar object in the right lower area of the field was replaced with a novel object. (**b**) Bar graphs depicting the quantification of NOR results. (**c**) Open Field testing suggests that CBD inhalation could ameliorate the cognitive function in 5xFAD mice. The figure shows a heat map generated from an Open Field test (OF) in 5xFAD mice treated with inhaled CBD and/or placebo. (**d**) OF results are quantified and displayed as bar graphs, suggesting improved cognitive function in AD mice treated by CBD inhalation compared to mice treated by placebo (** *p* ≤ 0.01).

**Figure 4 ijms-25-11764-f004:**
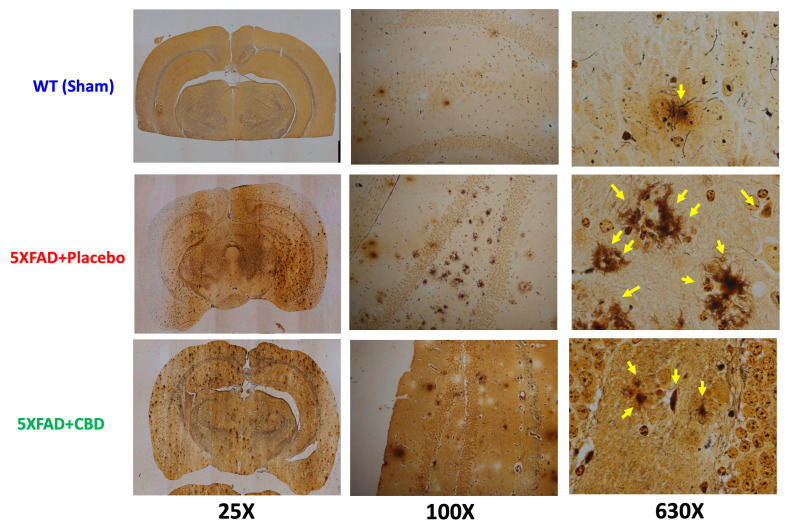
CBD inhalation reduced senile plaques in AD. Using the Bielschowsky staining, nerve fibers, neurofibrillary tangles, and senile plaques in the central nervous system were measured. Inhaled CBD significantly reduced senile plaques (amyloid plaques, yellow arrows) and neurofibrillary tangles (in black/dark brown) in AD mice compared to mice treated with placebo.

**Figure 5 ijms-25-11764-f005:**
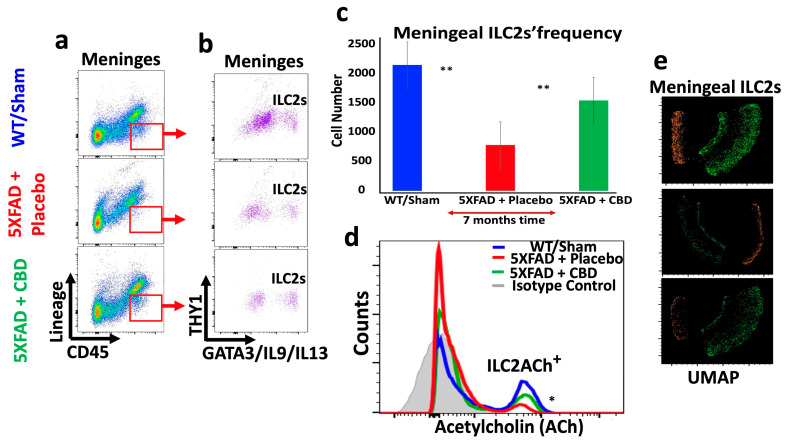
Inhaled CBD shifts immune profile in meninges of mice with AD towards the normal status. (**a**,**b**) Flow cytometry analysis showed depletion of meningeal ILC2s in 5xFAD mice with AD compared to the WT mice. Inhaled CBD reversed such alteration towards the normal status. (**c**) Quantification of meningeal ILC2s frequencies with significant differences between CBD-treated mice versus the placebo group (** *p* ≤ 0.02) as well as between normal WT and AD mice treated with placebo (** *p* ≤ 0.01). (**d**) Histograms from flow cytometric analysis revealed that CBD inhalation enhanced ACh expression by meningeal ILC2s significantly compared to the placebo treated group (* *p* ≤ 0.05). (**e**) UMAP analytical pattern demonstrated changes in the distribution and frequency of meningeal ILC2s in WT and 5xFAD mice treated with placebo/CBD.

**Figure 6 ijms-25-11764-f006:**
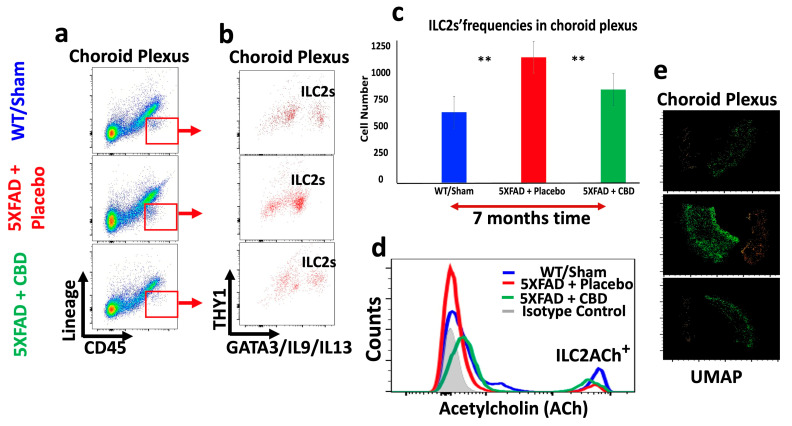
Inhaled CBD shifts immune profile in the choroid plexus of mice with AD towards the normal status. (**a**,**b**) Flow cytometry analysis showed excessive accumulation of ILC2s in the choroid plexus (CP) of 5xFAD mice with AD treated with placebo compared to AD mice with CBD inhalation. Inhaled CBD reversed such alteration towards the normal status, reduced ILC2s in the CP, and increased meningeal ILC2s. (**c**) Quantification of ILC2s frequencies in the CP with significant differences between CBD-treated mice versus the placebo group (** *p* ≤ 0.02) as well as between normal WT and AD mice treated with placebo (** *p* ≤ 0.02). (**d**) Histograms from flow cytometric analysis revealed that CBD inhalation enhanced ACh expression by ILC2s in the CP compared to the placebo-treated group. (**e**) UMAP analytical pattern demonstrated changes in the distribution and frequency of ILC2s in the CP of WT and 5xFAD mice treated with placebo/CBD.

## Data Availability

Original data are available upon request, please send requests to the corresponding author.

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
