# Peer review of "Boosting Acetylcholine Signaling by Cannabidiol in a Murine Model of Alzheimer’s Disease"

_ijms, 2024, doi:10.3390/ijms252111764_

Round 1

Reviewer 1 Report

Comments and Suggestions for Authors

The manuscript describes the potential health benefits of cannabidiol in AD based on the research conducted on a frequently used animal model. The main structure of the work should be improved to respect the quality imposed by the Journal, I would like to make some suggestions to improve the overall work:

1. I recommend you rewrite the abstract and make it structured (background; materials and method; results; conclusion)

2. In the introduction - the cholinergic hypothesis of AD is an older hypothesis, not a recent one as you state here. Please revise. Also, the last paragraph of the introduction could be improved.

3. Materials and methods: where is the ethical consent?? Where did you get the mice from? Subchapter on statistical analysis should be also improved. There are a lot of unanswered questions... 

4. The discussion section should also be improved. You could detail more on  the pathophysiology that explains cannabidiol's impact in AD. Also, the limitations of your study should be more developed and suggestions for future improvements/other studies to improve the topic.

5. The conclusion section should be improved. Also, future research directions should be mentioned.

Reviewer 2 Report

Comments and Suggestions for Authors

Using the 5xFAD mouse model, these workers have investigated whether cannabidiol (CBD) could potentially relieve Alzheimer’s disease (AD) symptoms by enhancing acetylcholine (ACh) production and slow its progression by reducing β-amyloid load. A second aim was to study CBD's ability to regulate the distribuiton of type 2 innate lymphoid cells (ILC2) which move from the meninges to the choroid plexus in AD and orchestrate inflammatory responses. To achieve this goal, ten five-month old 5xFAD mice were treated for seven months with 10 mg daily of broad-spectrum CBD spray which was inhaled through a mask while ten other 5xFAD mice inhaled hemp extract placebo. Mouse behaviour (open field, novel object recognition), histopathology, and ICL2 distribution were assessed and compared between these two groups and with wild type mice. CBD improved hippocampal ACh expression in the 5xFAD mice and their activity and recognition behaviour. Post mortem it had reduced both Aβ plaque and tau tangle load. Photoacoustic imaging detected higher cerebral blood flow in the CBD treated AD mice and a greater number of ICL2s were retained in the meninges. It is concluded that these findings provide a  rationale for further research to determine whether inhaled CBD may provide a low cost, non-invasive, efficacious therapy to manage AD in humans.

This is a novel and well performed study which suggests CBD may have both symptomatic  and neuroprotective benefits if taken early in AD. The mechanism by which Aβ plaque and tau tangle load is suppressed by CBD, however, remains unclear. Does CBD suppress intra neuronal synthesis of Aβ and/or its extracellular release or does it increase its glymphatic  clearance? While ILC2 cells orchestrate inflammatory responses these were not measured here. Was there any evidence that astrocytic and microglial activation was influenced by CBD? While the effects of CBD on ACh levels, Aβ plaque load, behaviour, and ILC2 distribution are intriguing none of these were normalised in the 5xFAD mice. The presence of raised CBF after CBD, while encouraging, does not tell us whether it was coupled to improved metabolism.  In summary, this report is interesting but raises a number of questions about how CBD acts. A problem in translating CBD for use in AD patients will be that its side effects include fatigue and drowsiness - currently it is only licensed for use in resistant epilepsy.

Round 2

Reviewer 1 Report

Comments and Suggestions for Authors

Dear Authors, 

Thank you for addressing the issues I have raised in my previous review. The quality of the article has improved and is now suitable to be published in the Journal.